# The Role of Cortisol in Chronic Stress, Neurodegenerative Diseases, and Psychological Disorders

**DOI:** 10.3390/cells12232726

**Published:** 2023-11-29

**Authors:** Emilija Knezevic, Katarina Nenic, Vladislav Milanovic, Nebojsa Nick Knezevic

**Affiliations:** 1Department of Anesthesiology, Advocate Illinois Masonic Medical Center, Chicago, IL 60657, USA; ekneze2@illinois.edu (E.K.); katarinanenic@knights.ucf.edu (K.N.); vmilan3@uic.edu (V.M.); 2College of Liberal Arts and Sciences, University of Illinois at Urbana-Champaign, Champaign, IL 61820, USA; 3Department of Psychology, University of Central Florida, Orlando, FL 32826, USA; 4College of Medicine Rockford, University of Illinois, Rockford, IL 61107, USA; 5Department of Anesthesiology, University of Illinois, Chicago, IL 60612, USA; 6Department of Surgery, University of Illinois, Chicago, IL 60612, USA

**Keywords:** cortisol, hypothalamic–pituitary–adrenal (HPA) axis, chronic stress, neuroinflammation, physiological processes

## Abstract

Cortisol, a critical glucocorticoid hormone produced by the adrenal glands, plays a pivotal role in various physiological processes. Its release is finely orchestrated by the suprachiasmatic nucleus, governing the circadian rhythm and activating the intricate hypothalamic–pituitary–adrenal (HPA) axis, a vital neuroendocrine system responsible for stress response and maintaining homeostasis. Disruptions in cortisol regulation due to chronic stress, disease, and aging have profound implications for multiple bodily systems. Animal models have been instrumental in elucidating these complex cortisol dynamics during stress, shedding light on the interplay between physiological, neuroendocrine, and immune factors in the stress response. These models have also revealed the impact of various stressors, including social hierarchies, highlighting the role of social factors in cortisol regulation. Moreover, chronic stress is closely linked to the progression of neurodegenerative diseases, like Alzheimer’s and Parkinson’s, driven by excessive cortisol production and HPA axis dysregulation, along with neuroinflammation in the central nervous system. The relationship between cortisol dysregulation and major depressive disorder is complex, characterized by HPA axis hyperactivity and chronic inflammation. Lastly, chronic pain is associated with abnormal cortisol patterns that heighten pain sensitivity and susceptibility. Understanding these multifaceted mechanisms and their effects is essential, as they offer insights into potential interventions to mitigate the detrimental consequences of chronic stress and cortisol dysregulation in these conditions.

## 1. Introduction

Chronic stress has long been examined as a crucial factor in the development and progression of illness. This manuscript delves into the multifaceted interplay between cortisol and various diseases by examining the immune, nervous, and endocrine systems, shedding light on the profound implications of cortisol and hypothalamic–pituitary–adrenal (HPA) axis dysregulation for the onset, progression, and potential prevention pathways of conditions such as depression, Alzheimer’s disease, and Parkinson’s disease. By exploring the molecular, neurobiological, and clinical dimensions of stress, we aim to contribute a comprehensive review of the existing literature on the role cortisol plays in illness with the ultimate goal of informing more effective prevention and therapeutic strategies as well as the direction for future research in this area.

## 2. Mechanisms

Cortisol, a glucocorticoid hormone produced by the adrenal glands, plays a crucial role in various physiological processes in the human body [1]. Cortisol release follows the circadian rhythm that is regulated by the pacemaker within the suprachiasmatic nucleus [2,3]. The nucleus activates the hypothalamic–pituitary–adrenal (HPA) axis. Its mechanisms are finely tuned to respond to stress and maintain homeostasis [4]. One key mechanism involves the hypothalamic–pituitary–adrenal (HPA) axis, a complex neuroendocrine system. When the body perceives stress, the hypothalamus releases a corticotropin-releasing hormone (CRH), which stimulates the anterior pituitary gland to produce an adrenocorticotropic hormone (ACTH). The ACTH, in turn, prompts the adrenal cortex to synthesize and release cortisol [5]. Approximately 5% of all cortisol circulates in a free, soluble form. The other 90–95% is bound to proteins, including 80% to corticosteroid-binding globulin (CBG) with high affinity and up to 15% to albumin with low affinity [6]. Out of the two forms, only free cortisol is a biologically active hormone that can enter cells and interact with glucocorticoid receptors, thus providing feedback inhibition in the hypothalamus and pituitary gland, which are ultimately responsible for crucial functions, such as controlling inflammation and ensuring euglycemia [7].

Over the past decade, significant progress has been made in understanding the regulation of the hypothalamic–pituitary–adrenal (HPA) axis. Homeostatic physiological circuits integrate various internal and external signals to produce an appropriate response for target tissues, and the HPA axis exemplifies a homeostatic system [8]. Recent research reveals that the circadian rhythm of adrenal glucocorticoid hormones, corticosterone in rodents and predominantly cortisol in humans, consists of pulses with varying amplitudes [9]. These pulses are generated by a sub-hypothalamic pulse generator. The oscillating endogenous glucocorticoid signals interact with regulatory systems within different parts of the HPA axis, including the adrenal gland itself, where a regulatory network can modify the pulsatile hormone release [10]. The HPA axis output is a dynamic, oscillating glucocorticoid signal that requires decoding at the cellular level [11]. Eliminating the pulsatile signal through long-acting synthetic glucocorticoid administration can disrupt physiological regulation and negatively affect various glucocorticoid-dependent bodily systems [12]. Even slight alterations in the system’s dynamics during chronic stress or specific disease states can potentially impact the functional output of multiple cells and tissues throughout the body. This can lead to changes in metabolic processes, behavior, mood, and cognitive function in susceptible individuals [13].

Mineralocorticoid receptors (MRs) and glucocorticoid receptors (GRs) are the two types of receptors that glucocorticoids exert their effects through on the hypothalamic–pituitary–adrenal axis (HPA axis) [14]. Whereas GRs are present throughout the CNS, MRs are mostly concentrated in the limbic structures, specifically the hippocampus [15]. MRs have a much higher affinity to glucocorticoids than GRs (high only at the peak of the circadian rhythm and immediately after exposure to stress), and studies on GC receptor occupancy have provided a basis for the proposition that MRs (proactive feedback mode) mediate the inhibitory action of the hippocampus on HPA activity, whereas GRs (reactive feedback mode) mediate HPA activity under conditions of elevated glucocorticoid levels [15,16]. The binding of cortisol to glucocorticoid receptors initiates a cascade of intracellular events, leading to the regulation of gene transcription and ultimately influencing various physiological processes. Cortisol helps regulate glucose metabolism by increasing gluconeogenesis and insulin resistance, ensuring an adequate energy supply during stress [17]. Cortisol also influences the cardiovascular system by regulating blood pressure and vascular tone. It enhances vasoconstriction and increases cardiac output, contributing to the body’s readiness for the “fight or flight” response during acute stress [18].

In the 1950s, when Hench, Kendall, and Reichstein demonstrated the powerful anti-inflammatory effects and applications of cortisone in clinical use to treat inflammatory diseases, such as rheumatoid arthritis, the impact of glucocorticoids on inflammation in the body and their regulation of it is clear [19]. Glucocorticoids play a powerful role in regulating inflammation, and their anti-inflammatory effect has been known and utilized in clinical applications for decades. However, exposure to chronic stress promotes sustained, non-resolving inflammation within the CNS through the gradual development of glucocorticoid resistance, which further drives inflammation. Social conflicts and stress activate the sympathetic nervous system (SNS) and HPA axis. The two act together and release norepinephrine (NE), which upregulates the transcription of proinflammatory immune response genes IL-1, IL-6, and TNF [20,21]. In the case of GC resistance due to chronic stress, immune cells demonstrate decreased sensitivity to glucocorticoids. Therefore, in this case, cortisol release does not result in significant anti-inflammatory effects, as one would otherwise expect. Once GC resistance develops, the “fight or flight” response to social threats is altered and results in exaggerated inflammation, consequently perpetuating the proinflammatory state [22].

As mentioned above, under normal conditions and short-term stress, the glucocorticoids released are implicated in a mechanism of inhibiting proinflammatory cytokines and upregulating anti-inflammatory cytokines. Interleukin-1 beta (IL-1 beta) seems to be the culprit of stress-induced adverse effects [23] that stimulate the release of norepinephrine, which further causes a cascade of events involving the central nervous, endocrine, and immune systems. This modulation of the immune response is crucial in preventing excessive inflammation but can also make individuals more susceptible to infections and impair wound healing under chronic stress conditions [24].

Chronic stress exerts its ill effects over time through a mechanism of prolonged cortisol release; consequently, the HPA axis seems to become increasingly desensitized over a period of time. One of the proposed mechanisms is that constant GC release will blunt HPA axis response over time, leading to HPA axis dysregulation and cortisol resistance, which are implicated in many different diseases like Alzheimer’s disease, Parkinson’s disease, depression, etc., through a plethora of different factors, most notably an interplay between the central nervous system, endocrine, and immune interactions.

It is proposed that in chronic stress, a resulting increase in the IL-1 beta proinflammatory cytokine is transformed into a nervous signal—norepinephrine (NE). Consequently, endocrine system hormones melatonin and cortisol are utilized to provide a counterbalance against IL-1 beta. However, the counterbalancing is unsuccessful due to chronic stress-induced cortisol resistance. Consequently, a loop effect is created that increments all mediators, thus creating an imbalance between the immune, endocrine, and central nervous systems (Figure 1). The elevated levels of IL-1 beta appear to occur when an individual is unable to cope and succumbs to a stressful event, thus showing sickness behavior symptoms [25].

## 3. Animal Models and Mechanisms

Animal models have played a crucial role in unraveling the intricate mechanisms underlying the regulation of cortisol levels during stress [26]. These models allow researchers to stimulate and manipulate stressors, offering valuable insights into the physiological and neuroendocrine responses associated with stress. By studying animals under controlled conditions, scientists can better understand the pathways and feedback systems that govern cortisol secretion [27].

One notable study by [28] employed a rodent model to examine the molecular mechanisms of cortisol synthesis within the adrenal glands during stress. Their research highlighted the role of the adrenocorticotropic hormone (ACTH) in driving cortisol production. Furthermore, it revealed the feedback loops involved in regulating cortisol levels, shedding light on the intricacies of HPA axis functioning.

In addition to shedding light on physiological mechanisms, animal models have also provided insights into how various stressors, from physical to psychosocial, influence cortisol secretion. For instance, a study by [29] employed a non-human primate model to explore the impact of social stress on cortisol dynamics. Their work demonstrated the significant influence of social hierarchies within primate groups on cortisol regulation. Dominant individuals exhibited distinct cortisol profiles, emphasizing the role of social factors in modulating cortisol secretion.

The main glucocorticoid for stress regulation in rodents is plasma corticosterone; however, it is uncertain whether plasma cortisol can serve as an indicator of rodent stress activation. A study by [30] examined the effects of the estrous cycle stage, circadian rhythm, and various stressors on serum cortisol and corticosterone in mice. The research found a strong correlation (r = 0.6–0.85) between serum cortisol and corticosterone across both conditions or predictable and unpredictable stress. Both hormones peaked on day 1 of repeated or unpredictable stress, but the dynamics differed afterward. The corticosterone declined during repeated restraints but remained high during unpredictable stress. During forced swimming or heat stress, cortisol peaked within 3 min, while corticosterone peaked after 40 min. The results conclude that corticosterone is a more adaptation-related biomarker during chronic stress, while cortisol responds more quickly during severe acute stress.

These animal models not only aid in uncovering the fundamental mechanisms governing cortisol levels during stress but also provide insights that have implications for human health [31]. By understanding the intricacies of cortisol regulation in response to stress, researchers can develop targeted interventions and treatments for stress-related disorders in both animals and humans [32].

## 4. Measurements

Guidelines for measuring salivary diurnal cortisol in interventional studies are currently lacking, posing challenges for incorporating salivary cortisol as a biomarker in randomized controlled trials (RCTs) for behavioral and health therapies. The inherent complexity of salivary diurnal cortisol as a biomarker introduces unique difficulties in RCTs, as different interventions may exert varying effects on the composite diurnal profile, consisting of the cortisol awakening response, diurnal slope, and area under the curve—each reflecting a distinct aspect of the HPA axis function. This complexity influences decisions regarding the main measurement parameter, theories about potential paths for change, and assessments of effectiveness, target engagement, and action mechanisms. Moreover, concerns have been raised about the biomarker’s long-term stability over spans exceeding one month and its short-term reliability in the context of day-to-day state effects, non-compliance, and other factors [33,34,35,36].

In the realm of stress research, monitoring quantified long-term stress markers is vital for preventing and intervening early in stress-related chronic diseases. While physiological parameters such as heart rate, blood pressure, and metabolic hormones quantify acute and chronic stress, determining cortisol levels’ specific temporal characteristics remains challenging. Serum and salivary cortisol levels indicate acute changes at a single time point, but their utility in assessing long-term cortisol exposure is hindered by circadian fluctuations and protein binding. Hair cortisol concentration (HCC), measured from hair samples, offers a promising method for retrospectively assessing chronic stress over extended periods. The predictable growth rate of scalp hair, approximately 1 cm per month, makes the 1 cm hair segment a reflection of the prior month’s cortisol production. Combining chronic HCC measurements with acute cortisol assessments provides a comprehensive understanding of the stress response and its relevance to psychiatric and stress-related disorders [37,38].

In some studies, diurnal cortisol serves as a comprehensive measure, capturing cortisol levels from morning to night and examining the diurnal cortisol slope, describing the behavior of cortisol levels over the course of the day. Additionally, studies may focus on the cortisol awakening response (CAR), measuring cortisol within the first hour after awakening, a period when cortisol typically rises before gradually declining throughout the day, with a nighttime increase. Integrating diverse cortisol measurement approaches contributes to a nuanced understanding of stress dynamics in various contexts.

## 5. Overall Illness and Stress-Related Disorders

Numerous studies emphasize the substantial impact of chronic stress on the development and exacerbation of chronic pain disorders, such as fibromyalgia and migraines [39]. Migraines, characterized by recurrent throbbing headaches often accompanied by nausea, light, and sound sensitivity, are notably influenced by chronic stress, affecting both the onset and duration of these episodes [40]. Sensitized meningeal afferents innervating the dural vasculature are implicated in migraine pathophysiology, with these afferents projecting to the nucleus caudalis in the trigeminal ganglion [41]. Trigeminal meningeal nociceptors become sensitive to sterile inflammation in the intracranial meninges and are triggered by various factors including nutrition, hormonal swings, chronic stress, or events, like cortical spreading depression [42]. Stress, considered a major factor exacerbating migraine pain, has been shown in preclinical rodent studies to cause the degranulation of intracranial mast cells, an effect mediated by neuropeptides [43].

Similarly, a robust correlation has been established between chronic stress and fibromyalgia, a complex condition characterized by widespread musculoskeletal pain, fatigue, and localized soreness. The intricate neurobiological interplay between stress and these chronic pain syndromes involves alterations in pain processing pathways, the modulation of the immune system, and the dysregulation of hormones. Recognizing the role of chronic stress in migraines and fibromyalgia advances our understanding of the etiological factors contributing to these conditions, underscoring the importance of comprehensive, holistic management approaches that consider both physiological and psychosocial aspects. Fibromyalgia is characterized by chronic widespread pain (CWP), which significantly impacts the lives of those affected [44,45,46,47]. The management of FM is challenging, with limited effectiveness shown by pharmacological therapies [48,49,50,51]. Promising outcomes in terms of prevention are observed when regional and broad pain is diagnosed early and treated appropriately, with prominent risk factors including somatic symptoms, sick behavior, regional or non-widespread pain, and sleep disturbance [52,53,54]. Steroidal hormone cortisol, a marker of HPA axis dysfunction, reflects changes in FM and CWP.

Stress is widely acknowledged for its pivotal role in the development of mental illnesses and stress-related disorders, yet the intricate biological mechanisms governing pathological stress regulation continue to challenge our comprehension [55]. Recent research has made significant strides in shedding light on these mechanisms [56], notably through the innovative use of hair cortisol concentration (HCC) measurements. A noteworthy study, “Hair cortisol concentration and common mental disorder: A population-based multi-cohort study” [57], underscores the relevance of this approach. The study unveils a compelling association between elevated HCC and exposure to stressful life events [58], providing tangible evidence of the link between chronic stress and physiological markers. By quantifying cortisol levels in hair samples, this research not only highlights the profound impact of stress on the body but also emphasizes the potential of HCC as a robust biomarker for assessing stress-related psychiatric conditions [59]. This innovative method holds promise in advancing our understanding of the intricate interplay between stress, cortisol regulation, and the development of mental health disorders, potentially paving the way for more precise diagnostic and therapeutic strategies in the realm of mental health [60].

Stress management techniques, including relaxation exercises, mindfulness, and cognitive behavioral therapy (CBT), have emerged as valuable interventions for individuals seeking to regain control over their stress response and alleviate pain [61]. Several studies have underscored their efficacy in mitigating the physiological and psychological impact of stress [62]. For instance, mindfulness-based stress reduction (MBSR) has been shown to reduce cortisol levels, enhance emotional regulation, and improve pain perception [63]. Similarly, CBT, with its focus on modifying maladaptive thought patterns and behaviors, has been associated with reductions in stress-related symptoms and pain intensity [64]. Relaxation exercises, such as progressive muscle relaxation and deep breathing, contribute to decreased muscle tension and heightened relaxation responses, which can alleviate stress-related physical discomfort [65]. These stress management techniques provide valuable tools for individuals seeking to address stress, highlighting their potential to improve both physical and mental health outcomes [62].

## 6. Alzheimer’s and Parkinson’s Diseases

Stress and aging are prerequisites for neurodegenerative disease [66]. Although glucocorticoids are essential to homeostasis and a normal response to stress, excessive glucocorticoid production under conditions of chronic stress is implicated in many disease models [67]. The overproduction of GCs and a dysfunctional HPA axis keep appearing in the literature addressing possible contributions to the processes that are involved in the development of Alzheimer’s disease [68,69,70]. There seems to be a cyclical relationship between key brain areas, such as the hippocampus, its atrophy, and the feedback on the HPA axis contributing to its dysfunction [71] (Figure 2). Normally, the hippocampus exerts an inhibitory effect on the HPA axis. One of the proposed mechanisms is that since the hippocampus is one of the primary areas compromised in AD and contains the highest number of GC receptors in the brain, it is especially susceptible to the detriment of chronically excessive GC production [72]. Neurodegeneration occurs in critical brain areas, such as the hippocampus and substantia nigra (two of the crucial brain areas implicated in AD and PD, respectively), and leads to an increase in oxidative stress, mitochondrial dysfunction, and metabolic changes [73].

In recent years, researchers have used the term “type 3 diabetes”, first proposed over a decade ago, [74] to refer to Alzheimer’s disease due to the plethora of shared mechanisms implicated both in AD and type 1 and type 2 diabetes mellitus (T1DM and T2DM) [75]. In T2DM, one of the markers also involved in AD is inflammation that is caused due to insulin resistance in diabetes, which in turn causes a surge in proinflammatory cytokines (IL-6, IL-1 beta) and tumor necrosis factor-alpha (TNF-α) [76,77,78]. This is the same mechanism that is triggered by chronic stress in AD and which we delve into below in the section on AD and PD. Increased oxidative stress and mitochondrial dysfunction, which are mentioned above and occur in critical brain areas to cause neurodegeneration implicated in diseases, such as AD and PD [73], occur due to insulin resistance in T2DM [75]. It might be important to acknowledge the similar inflammatory mechanisms implicated in T2DM for future work examining chronic stress and their role in diabetes in more depth.

It is hard to conclude which comes first, the hypersecretion of GCs and the deleterious effect on neurons leading to atrophy of the hippocampus and subsequent dysregulation of the HPA axis given the negative feedback of the hippocampus being disrupted or the compromised hippocampus in AD, which leads to HPA axis dysfunction, causing GC hypersecretion to further contribute to neurodegenerative processes. One study found that in mice models of AD, heightened stress and glucocorticoid production aided in the production of beta-amyloid, a key feature of AD, and worsening cognitive and memory deficits [79]. In mice research, the administration of glucocorticoid receptor antagonists resulted in improved cognition and reduced AD pathologies, supporting findings that glucocorticoid signaling influences and likely plays a contributing role in the development and progression of AD pathologies. Further research in mice showed that glucocorticoids reduce tau degradation, resulting in tau accumulation [80].

Another insightful study with rats was conducted in order to study the effects and mediation of chronic stress in the pathology of the hippocampal cytoskeleton [81]. The findings support the notion that chronic stress may in fact play a crucial role in either buffering against or making more susceptible to neurological injury and subsequent neuron loss. In this study, one of the markers of neurodegeneration worsened by stress was tau antigenicity, a major hallmark of AD. In rats exposed to stress, stress acted as a potentiator of excitotoxin-induced tau immunoreactivity accumulation contributing to neuron loss in the hippocampus [81].

Research in animals showed that chronic stress and cortisol release increase vulnerability to Alzheimer’s disease via the accumulation of tau and beta-amyloid, maladaptive immune responses, brain atrophy, and synaptic dysregulation [80]. Glucocorticoid receptor DNA binding sites exist in two genes associated with the formation of beta-amyloid (amyloid precursor protein and beta-site amyloid precursor protein cleaving enzyme 1). Therefore, it is thought that glucocorticoids can bind to these sites and influence beta-amyloid production [82]. In rat research, the administration of a glucocorticoid receptor antagonist resulted in improved cognition and reduced AD pathologies [83]. This evidence indicates that glucocorticoid signaling influences the likelihood and the extent of AD pathologies. Further research showed that glucocorticoids reduce tau degradation, resulting in tau accumulation [84].

Research demonstrated that microglia are important in forming memories and supporting neurogenesis within the brain [85,86]. Early in the AD process, microglia can phagocytose and clear accumulating beta-amyloids. However, with chronic stress, microglial response becomes ineffective with compromised phagocytosis. This results in a chronic proinflammatory state, resulting in an increasingly neurotoxic environment through the production of proinflammatory cytokines, altered synaptic pruning, and the reduced production of protective factors [87]. The research found apolipoprotein E transcripts to be significantly upregulated in AD-associated microglia [88,89]. Additionally, it was discovered that chronically activated microglia induce cognitive decline indirectly via the accumulation of neurofibrillary tangles [90].

Human research indicates that microglia promote beta-amyloids and subsequent tau accumulation, thus having a key role in cognitive decline [90,91]. The nucleotide-binding oligomerization domain-like receptor, pyrin domain-containing 3 (NLRP3), within microglia recruits the adaptor protein apoptosis-associated speck-like protein containing a C-terminal caspase recruitment domain (ASC) to form an inflammasome complex. This inflammasome complex is responsible for the production of IL-1 beta, which is a proinflammatory cytokine. Both beta-amyloids and tau can activate the NLRP3-ASC inflammasome within microglia [92,93]. Rat research has identified the NLRP3-ASC inflammasome as a major mechanism through which microglia may drive pathological features of AD [92,93,94]. Furthermore, it was discovered that beta-amyloid oligomers have the ability to facilitate NLRP3 inflammasome activation in microglia cells, indicating that inflammation in the CNS may precede the accumulation of beta-amyloid plaques [95]. The inflammasome could have an independent role in promoting tau aggregation. The disruption of the NLRP3-ASC inflammasome in mice resulted in significantly decreased forming of tau aggregates [93]. The NLRP3-ASC inflammasome mechanisms contributing to increased beta-amyloids and tau are thought to be a combination of kinases and phosphatases involved in tau phosphorylation [96] through the production of IL-1 beta. IL-1 beta was shown to exacerbate tau pathologies [97] and reduce microglial phagocytic activity [92,93,94]. Ultimately, chronic stress is associated with AD progression by priming microglia to enter an increased proinflammatory response and drive the accumulation of phosphorylated tau following exposure to a secondary stimulus, such as an accumulating beta-amyloid [80].

Parkinson’s disease is a progressive neurodegeneration of the dopaminergic neurons in the substantia nigra. Chronic stress and elevated cortisol are implicated in many of the mechanisms that are implicated in the progression and, potentially, development of the disease including metabolic changes, mitochondrial dysfunction, and neuroinflammation [78] (see Figure 3 for a possible disease progression/development flowchart). Normal cortisol secretion acts inhibitory toward cells that produce peripheral cytokines, mitigating inflammation. On the other hand, excessive cortisol secretion promotes neuroinflammation through the mechanism in which glucocorticoid receptors or GR (anti-inflammatory actions) are downregulated as a compensatory response to excess cortisol production and GC resistance, which further promote the release of GCs. Under normal conditions, catecholamines and glucocorticoids regulate inflammation through the inhibition of proinflammatory cytokines, such as IL-6, IL-1*β*, and TNF-*α*, and the stimulation of anti-inflammatory cytokines, such as IL-4, IL-10, and IL-13 [38], by exerting action in the cytoplasm of immune cells. Through an interplay of inflammation due to immune response and subsequent cytokine release and a lack of downregulation normally executed by GCs, neuroinflammation persists, leading to the atrophy of neurons in critical brain areas, such as the hippocampus [79,98]. Neuroinflammation is prominent in diseases, such as Parkinson’s disease and Alzheimer’s disease, and may be the driving factor in both the progression of and contribution to disease development. When examining the proinflammatory cytokines in PD, such as IL-6 and IL-1 beta, researchers have observed an increase in these inflammation-promoting cytokines in the diseased brains of both AD and PD sufferers [99,100], which could suggest the crucial role of inflammation exacerbated by chronic stress in neurodegenerative disease.

Aging, as the crucial risk factor for PD development coupled with chronic stress contribute to the increased cortisol levels, and vice versa, which together drive the mechanisms (oxidative stress, neuro-inflammation, mitochondrial dysfunction, more ROS, and metabolic changes) that triggers the development and the progression of PD.

## 7. Depression

Experiencing acute stress induces the temporary release of cortisol in increments that eventually decrease. This temporary incremental release is not dangerous for long-term health outcomes. However, research has shown that during chronic stress, cortisol loses its circadian rhythm [23]. This results in GC resistance due to the desensitization of GR and the absence of a proper response to cortisol. Consequently, it is hypothesized that HPA axis hyperactivity and resulting GC resistance may represent a connection between chronic stress and a major depressive disorder [101,102].

Exposure to chronic stress promotes sustained, non-resolving inflammation within the CNS that could result in symptoms of depression [103]. Animal models of depression-like behavior demonstrated elevated proinflammatory cytokine levels, most commonly IL-1 beta and tumor necrosis factor-alpha (TNF-alpha) [104,105] (see Table 1). IL-1 beta promotes proinflammatory genes and proteins in the brain [106,107,108], as well as IL-1 beta-induced sickness behavior in rodents, resulting in social withdrawal, loss of appetite, decreased motor activity, and a decrease in cognitive function [109] (see Table 1). As supporting evidence, research demonstrated that the inhibition of the IL-1 beta receptor was able to rescue anhedonia in rats exposed to chronic stress [110] (see Table 1), thus demonstrating the importance of IL-1 beta in studying chronic stress-related depression. Furthermore, IL-1 beta exhibited the ability to activate the HPA axis and suppress the hypothalamic–pituitary–gonadal (HPG) axis [111,112,113]. IL-1 beta also stimulates NE release through its influence on the HPA axis [23]. In healthy subjects, cortisol decreases IL-1 beta, generating an anti-inflammatory response. However, with adaptations to chronic stress, cortisol’s function is diminished, which results in increased levels of IL-1 beta. Research showed that IL-1 beta can be produced by glial cells, neurons, and the immune system and that it can pass through the blood–brain barrier, resulting in sickness behavior symptoms [25]. These mechanisms and implications, which are mediated by chronic stress, could shed light on the possible etiology of depression through a complex interplay of the immune, endocrine, and nervous systems.

One large cohort study examined the implications of the hypothalamic–pituitary–adrenal axis in MDD. Researchers looked at individuals with both remitted and current MDD, as well as healthy controls. Their findings showed that both currently depressed and those with remitted MDD had a significantly higher cortisol awakening response (CAR) than their non-depressed counterparts [114]. Although the findings were modest, they were significant for both groups. Another study found similar associations: women suffering currently from depression had significantly higher average cortisol levels throughout the day, as well as a significantly higher CAR, than the non-depressed group [115]. A third study showed similarly intriguing findings when examining basal cortisol in various DSM-IV dimensions and categories of depression and anxiety [116]. The findings showed that CAR had a nonlinear relationship with anhedonic depression and general distress dimensions of the MASQ (specifically an inverted U-shape association). The indication here is that depressive symptoms are associated with both a hyper- and hypoactive HPA axis. The findings might suggest that disorders, like anxiety and depression, may initially, given high cortisol in the morning, cause the HPA axis to be hyperactive and with progression lead to a blunting of that response, yielding lower cortisol in the morning [58]. They also observed that diurnal cortisol decline had a linear relationship with general distress and anhedonic depression, pointing to an increased concentration of cortisol as the severity of symptoms increased. Patients with DSM-IV depressive disorder had higher concentrations of cortisol and a steeper slope than the controls [25]. The findings for studies measuring diurnal cortisol and CAR in MDD have been inconsistent. There are those that have demonstrated significant differences, demonstrating those with MDD to have higher measures than their non-depressed counterparts, like the ones mentioned above. On the other hand, there are also those that have not found a significantly hyperactive HPA axis in depression [25]. It is important to note that the methods of measuring cortisol vary extensively in their sensitivity to HPA axis function, and there seem to be differences that depend on the dimension and category of depression being examined [116].

Research suggests that the severity of depression is associated with the level of circulating cytokines and chemokines, which compromise the blood–brain barrier and activate the central immune response [117] (see Table 1). When the overactivated immune response exceeds the resolving capacity of resident immune cells within the brain, psychiatric disorders tend to develop. Peripheral CD4+ T cells appear to be major contributors to the occurrence of mental disorders [118,119] (see Table 1). Chronic stress induces the release of cytokines and promotes the differentiation of peripheral CD4+ cells into different phenotypes [120]. Th17 cells are particularly interesting because of their high pathogenic potential in CNS diseases [121,122]. Research showed an increase in Th17 cells in mice exhibiting learned helplessness. Additionally, mice receiving Th17 cells displayed significant depressive-like behaviors [119,123]. Multiple studies support the idea that the severity of depression is associated with both increased levels of proinflammatory cytokines and with structural as well as functional alterations in the dorsal part of the striatum [124,125,126]. Aberrant activity in the dorsal striatum is implicated in the core symptoms of depression, such as anhedonia and psychomotor retardation. Stressed rats exhibited significant symptoms of depression, blood–brain barrier disruption, and neuroinflammation in the dorsal striatum [117]. Research in rats conducted by Peng et al. showed a time-dependent increase in thymus-derived and spleen-derived naïve CD4+ T cells, along with aggregation of inflammatory Th17 cells in the dorsal striatum during exposure to stress. It was demonstrated that an increase in Th17-derived cytokines within the brain can impair the blood–brain barrier integrity. This facilitates CNS access for other immune cells and cytokines. Inflammatory cytokines, IL-1 beta and IL-6, were significantly elevated in the serum and dorsolateral striatum of rats exposed to chronic stress. In addition, IL-1 beta, IL-6, and TNF-alpha levels were elevated within the dorsomedial striatum. Furthermore, reactive astrogliosis was noted in the dorsolateral and dorsomedial striatum of rats exposed to chronic stress [117].

Notably, IL-17 and IL-22 are major Th17-produced cytokines that generate tissue inflammation [121]. IL-17 impairs blood–brain barrier integrity, allowing for more immune cells to cross into the CNS [127,128]. Astrocytes can respond to IL-17 and release mediators that promote tissue damage [129]. Recruitment of activated Th17 cells into the CNS and the increased production of IL-17 and IL-22 may be crucial for the development of symptoms of depression [117]. Importantly, Peng et al. showed that preventing the transformation of CD4+ T cells into inflammatory Th17 cells in the early phases of stress decreases stress-induced symptoms of depression. This prevention resulted in lower levels of expressed proinflammatory cytokines IL-6, IL-17, and IL-22.

**Table 1 cells-12-02726-t001:** Animal studies.

Study	Subjects	Mechanism of Stress	Measure of the Outcome	Results and Findings
Grippo et al. (2005) [104]	Male Sprague–Dawley rats	Exposure to anhedonia-inducing CMS (Chronic Mild Stress)	Anhedonia, defined as a reduction in sucrose intake without concomitant effect on water intake	Humoral assays showed increased levels of TNF-alpha and IL-1 beta
Hodes et al. (2014) [105]	C57BL/6J mice	Exposure to the social stress model: repeated social defeat stress due to exposure to a larger, aggressive CD-1 mouse, resulting in the development of depressive-like symptoms	Social avoidance and IL-6 levels	Elevated IL-6 levels in the serum of the stress-susceptible mice along with increased social avoidance behavior; IL-6 levels were strongly negatively correlated with social interaction behavior
Dantzer et al. (2008) [109]	Rats and mice	Administration of IL-1 beta or TNF-alpha	Behavior	Sickness behavior and depressive symptoms; rodents showed no interest in their physical and social environment, had decreased motor activity, exhibited social withdrawal, decreased food and water intake, and impaired cognition
Koo et al. (2008) [110]	Control group: wild type rats Experimental group: rats with a blockade of an IL-1 beta receptor (IL-1RI) by an inhibitor or IL-RI null rats	Exposure to CUS (Chronic Unpredictable Stress)	Anhedonia, measured through sucrose preference testing	The control group exhibited anhedonia; in the experimental group, blockade of the IL-1 beta receptor blocked the anti-neurogenic effect of stress and prevented anhedonic behavior
Peng et al. (2022) [117]	Rats	6 h of daily CRS (Chronic Restraint Stress)	Behavior, BBB changes, neuroinflammation, and CD4+ T cell level measurements	Rats exhibited depressive-like symptoms, BBB disruption, and neuroinflammation in the dorsal striatum. There was also a time-dependent increase in thymus- and spleen-derived CD4+ T cells. Inhibition of CD4+ cell differentiation with SR1001 in the early stages of CRS exposure ameliorated stress-induced depressive-like behavior and the inflammatory response
Beurel et al. (2018) [119]	Male Rag2 knockout mice	Administration of Th17, Th1, and Treg cells	Behavior	Administration of Th17, but not Th1 or Treg cells, increased susceptibility to learned helplessness depressive-like behavior

## 8. Pain

Chronic pain is a debilitating condition that affects millions of individuals worldwide, and it has been associated with alterations in cortisol levels [1]. It is often associated with a cascade of physiological and psychological responses, one of which is altered cortisol levels [130]. Several studies have explored the dysregulation of the hypothalamic–pituitary–adrenal (HPA) axis in individuals with chronic pain, as well as the crucial role of the glucocorticoid receptor (GCC) [130] (see Figure 4). The HPA axis is a key regulator of cortisol production, and abnormalities in the axis can lead to alterations in cortisol levels [24].

This dysregulation can manifest as elevated baseline cortisol levels and blunted diurnal cortisol patterns in individuals with chronic pain conditions, such as fibromyalgia [131] and chronic low back pain [132]. Additionally, there is evidence to suggest that cortisol dysregulation may exacerbate the experience of chronic pain [133]. Chronic stress, often associated with HPA axis dysregulation, can contribute to increased pain sensitivity and the development of hyperalgesia, making individuals more susceptible to pain [67]. Moreover, elevated cortisol levels in response to chronic pain may perpetuate the pain cycle by contributing to muscle tension and inflammation [134].

The glucocorticoid receptor (GCC), a crucial component of the HPA axis, plays a significant role in the regulation of chronic pain [130]. The GCC is essential for modulating the anti-inflammatory and immunosuppressive actions of cortisol [135]. Dysfunctional GCC signaling is implicated in chronic pain disorders, such as fibromyalgia and neuropathic pain, where alterations in GCC expression or sensitivity may influence the body’s ability to control inflammation and modulate the nociceptive response [135].

Stress adaptation encompasses the hypothalamic–pituitary–adrenocortical (HPA) axis, where glucocorticoids are released upon activation to reallocate energy resources [136]. Corticotropin-releasing hormone (CRH) production in the hypothalamic paraventricular nucleus (PVN) constitutes a key neuronal mechanism underlying HPA stress response [136]. Chronic stress can activate the HPA axis through heightened stress responses, persistent basal hypersecretion, and adrenal depletion, with the manifestation of this reaction dependent on various factors. Distinct brain mechanisms may govern chronic stress responses compared to acute ones, involving the activation of new limbic, hypothalamic, and brainstem circuits [137]. Whether an individual’s acute or chronic stress reactions are maladaptive or adaptive depends on factors such as age, sex, early life experiences, environment, and heredity.

Understanding the intricate interplay between the HPA axis, glucocorticoid signaling, and stress responses underscores the need for a comprehensive approach to chronic pain management that considers the multifaceted nature of stress-induced alterations in the body’s regulatory systems [130]. Investigating the intricate interactions among chronic stress, the GCC/HPA axis, and chronic pain holds promise for identifying novel targets in pain treatment. The intricate relationship between cortisol and chronic pain is crucial for developing effective pain management strategies [130].

## 9. Conclusions

In conclusion, cortisol, a pivotal glucocorticoid hormone produced by the adrenal glands, is integral to numerous physiological processes. Its release is intricately regulated by the suprachiasmatic nucleus, governing the circadian rhythm and activating the complex hypothalamic–pituitary–adrenal (HPA) axis, a vital neuroendocrine system responsible for stress response and maintaining homeostasis. Disruptions in cortisol regulation due to chronic stress, disease, and aging have profound implications for multiple bodily systems. Animal models have been essential in unraveling these complex cortisol dynamics during stress, shedding light on the interplay between physiological, neuroendocrine, and immune factors in the stress response. These models have also revealed the impact of various stressors, including social hierarchies, highlighting the role of social factors in cortisol regulation. There is a big spectrum of psychosomatic diseases that are a result of chronic stress; however, in this review, we focused only on the most frequent diseases that affect the central nervous system. Chronic stress is closely linked to the progression of diseases, like Alzheimer’s and Parkinson’s, driven by excessive cortisol production and HPA axis dysregulation, along with neuroinflammation in the central nervous system. The relationship between cortisol dysregulation and major depressive disorder is complex, characterized by HPA axis hyperactivity and chronic inflammation. Lastly, chronic pain is associated with abnormal cortisol patterns that heighten pain sensitivity and susceptibility and are closely related to depression. Understanding these multifaceted mechanisms and their effects is essential, as they offer insights into potential interventions to mitigate the detrimental consequences of chronic stress and cortisol dysregulation in these conditions.

## Figures and Tables

**Figure 1 cells-12-02726-f001:**
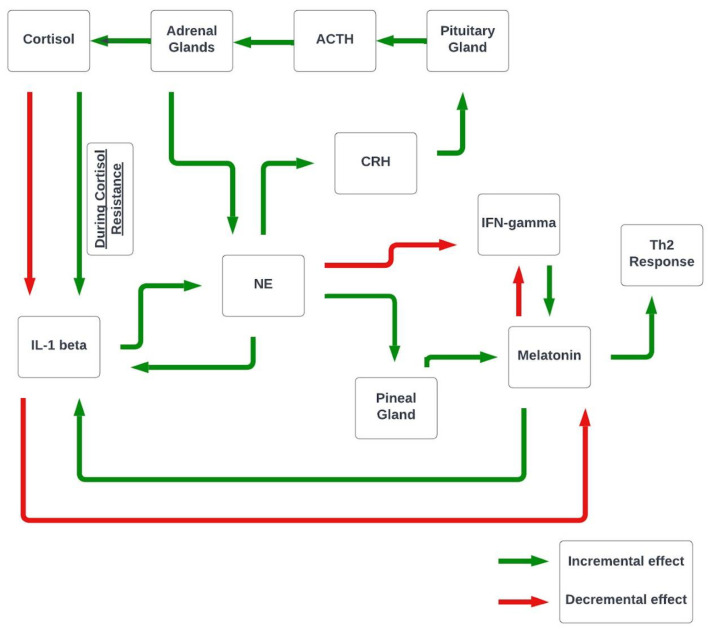
Proposed interplay between the CNS, endocrine, and immune systems.

**Figure 2 cells-12-02726-f002:**
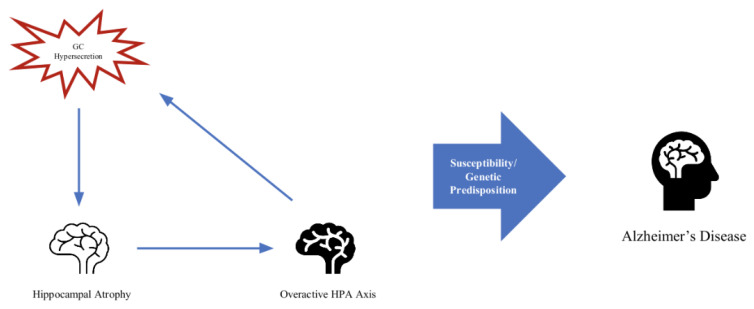
Feedback loop of hippocampal atrophy, overactive HPA axis, and GC hypersecretion.

**Figure 3 cells-12-02726-f003:**
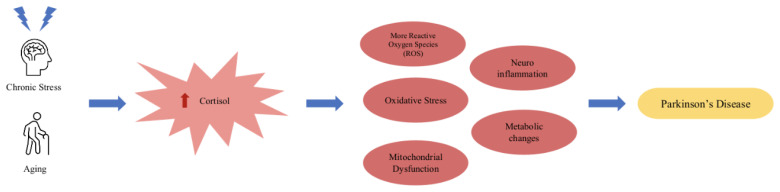
Model for the potential mechanism of cortisol dysregulation implication in PD.

**Figure 4 cells-12-02726-f004:**
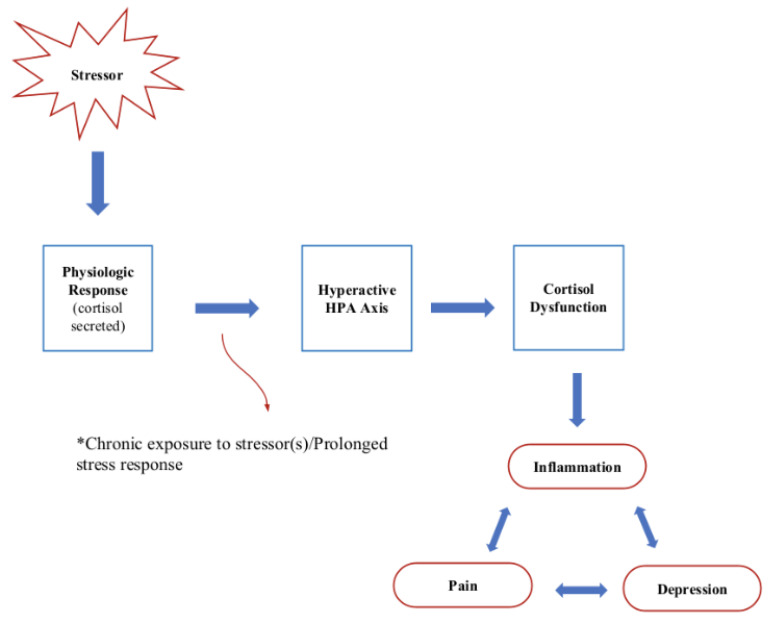
Chronic stress mechanism and implications.

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
