# Peer review of "The Role of Cortisol in Chronic Stress, Neurodegenerative Diseases, and Psychological Disorders"

_cells, 2023, doi:10.3390/cells12232726_

Round 1

Reviewer 1 Report

Comments and Suggestions for Authors

Literature data related to the role of cortisol in chronic stress, neurodegenerative diseases, and psychological disorders were presented in this in-depth review paper. The review is written in an easy-to-understand style and addresses a timely topic of considerable relevance. Insights into potential therapies to reduce the adverse effects of chronic stress and cortisol dysregulation in these circumstances can be gained by a thorough understanding of the complex mechanisms in action here.

Comments:

1. Since animal models have played a crucial role in unravelling the intricate mechanisms underlying the regulation of cortisol levels during stress, the authors may present a summary schematic representation or table of animal models and mechanisms (Section 3), which could help readers better understand the significance of the listed models and especially mechanisms.

2. Since Alzheimer's disease is associated with insulin resistance, a feature of type 2 diabetes, the term "type 3 diabetes" has been used to refer to the neurodegenerative disorder in recent years. In a few sentences, the authors should present recent literature data regarding Alzheimer's disease classified as Type 2 Diabetes.?

3.      Besides measuring cortisol levels in the hair samples, the authors mentioned measurements of diurnal cortisol levels. How about measuring cortisol levels in saliva?

4. The authors mentioned the correlation between chronic stress and chronic pain in general. Can authors please emphasize the role of chronic stress in two widespread chronic pain conditions that are well-known to be stress-related, such as migraines and fibromyalgia?

Author Response

  1. Since animal models have played a crucial role in unravelling the intricate mechanisms underlying the regulation of cortisol levels during stress, the authors may present a summary schematic representation or table of animal models and mechanisms (Section 3), which could help readers better understand the significance of the listed models and especially mechanisms.

Dear Reviewer, thank you kindly for pointing this out. We have added a Table summarizing several animal studies, along with the modes of stress and research outcomes in Section 3, as suggested.

  1. Since Alzheimer’s disease is associated with insulin resistance, a feature of type 2 diabetes, the term “type 3 diabetes” has been used to refer to the neurodegenerative disorder in recent years. In a few sentences, the authors should present recent literature data regarding Alzheimer’s disease classified as Type 2 Diabetes?

Thank you for your comments and suggestions which have given us further insight for revisions/additions for our review. We have examined some recent literature that refers to AD as type 3 diabetes and have included some information on this in section 6 on AD and PD. The addition is included in the 2nd paragraph under this section.

  1. Besides measuring cortisol levels in the hair samples, the authors mentioned measurements of diurnal cortisol levels. How about measuring cortisol levels in saliva?

Thank you so much for pointing this out as there really is a lack and necessity of accurate biomarkers that can serve to measure stress. We have added information from a few studies describing the measurement of salivary diurnal cortisol levels and their measurement in RCTs and other studies.

  1. The authors mentioned the correlation between chronic stress and chronic pain in general. Can authors please emphasize the role of chronic stress in two widespread chronic pain conditions that are well-known to be stress-related, such as migraines and fibromyalgia?

Thank you so much for drawing our attention to two very important and well-known chronic pain conditions that we did not initially include. The interplay between those conditions and chronic stress is very valuable to know. We have included more information about migraines and fibromyalgia in our manuscript.

Reviewer 2 Report

Comments and Suggestions for Authors

The presented manuscript gives an interesting and updated overview about general mechanisms regarding stress and disease.

I have few, rather broad, suggestions/comments:

·       Cortisol also binds to mineralocorticoid receptors, not only to the GR. I would recommend to mention both receptor types and discuss potentially differential role in glucocorticoid resistance.

·        How did you decide the diseases/states mentioned? The circuit between GCC and inflammatory markers could theoretically explain any other kind of disease progression, e.g.: autoimmune diseases. I would suggest an overarching statement about those aspects that hold true for disease progression in the most general view.

·        As the review gives a broad overview, I would recommend to also briefly explain the HPA axis (CRH à ACTH à adrenals) as well as free versus bounded cortisol.

·        Due to my personal understanding, the most innovative aspect of your review relates to the association between GCC/HPA axis and chronic pain. Indeed, this topic falls rather short. Your review might improve by going more in depth in this regard.

Author Response

The presented manuscript gives an interesting and updated overview about general mechanisms regarding stress and disease. I have few, rather broad, suggestions/comments:

  • Cortisol also binds to mineralocorticoid receptors, not only to the GR. I would recommend to mention both receptor types and discuss potentially differential role in glucocorticoid resistance.

Dear Reviewer, thank you for your insight and comments and suggestions for our review. We have utilized your helpful feedback to improve on and add relevant information in order to enrich our paper. We added information on mineralocorticoid receptors and glucocorticoid receptors (differentiation and role in feedback on HPA axis) in the third paragraph under subtitle number 2 titled “Mechanisms” to include the role of both receptor types.

  • How did you decide the diseases/states mentioned? The circuit between GCC and inflammatory markers could theoretically explain any other kind of disease progression, e.g.: autoimmune diseases. I would suggest an overarching statement about those aspects that hold true for disease progression in the most general view.

We agree with you that GCC and inflammatory markers could explain a lot of psychosomatic disorders and unfortunately, due to a lack of space, we could only focus on a few of them. Our goal was to have a focus on the Central Nervous System and that is the reason why we chose some most frequent neurodegenerative and psychological disorders, such as Alzheimer’s, Parkinson’s, and Depression.

  • As the review gives a broad overview, I would recommend to also briefly explain the HPA axis (CRH à ACTH à adrenals) as well as free versus bounded cortisol.

Thank you kindly for bringing this to our attention. This edit would certainly be very useful to the readers and provide for a better read. Therefore, we added this information on the second page of our paper, under “Mechanisms” chapter, along with corresponding references.

  • Due to my personal understanding, the most innovative aspect of your review relates to the association between GCC/HPA axis and chronic pain. Indeed, this topic falls rather short. Your review might improve by going more in depth in this regard.

Thank you so much for your insight, we agree that this topic fell a little short in our initial manuscript. Therefore, we have expanded a bit more on the association between the GCC, HPA axis, and chronic pain in section 7. We also mentioned migraines and fibromyalgia as the most frequent disorders that are associated with chronic stress. However, this could be a topic for a separate comprehensive review.